# Mycotic Diseases in Chelonians

**DOI:** 10.3390/jof9050518

**Published:** 2023-04-27

**Authors:** Simona Nardoni, Francesca Mancianti

**Affiliations:** Dipartimento di Scienze Veterinarie, Università degli Studi di Pisa, 56124 Pisa, Italy

**Keywords:** chelonians, mycoses, turtles, tortoises, emerging mycoses, shell mycoses, skin mycoses, deep mycoses, treatment

## Abstract

Turtles and ‘tortoises’ populations are declining in number, the factors driving extinction risks being related to habitat loss and degradation, climate change, introduction of invasive plant and animal species, consumption by humans for food and medicinal use, and collection for international pet trade. Fungal infections represent one of the main threats for ecosystem health. The present narrative review deals with conventional and emerging mycoses of Chelonians. Although conventional mycoses in captive and pet reptiles would depend on poor husbandry, being the agents mostly opportunistic pathogens, some fungal species were reported to occur more frequently, such as the entomopathogen *Purpureocillium lilacinum*. Furthermore, emerging agents such as the *Fusarium solani* species complex have been recognized as a real threat for the surviving of some aquatic species, acting as primary pathogens. This complex has been recently included within pathogens in One Health issues. *Emydomyces testavorans* is recognized as an emerging threat, although, due its recent identification, information about its epidemiology is limited. Data about treatments and outcomes of mycoses in Chelonians are also referred.

## 1. Introduction

Turtles, terrapins, and tortoises belong to the order Testudines and have present on our planet for hundreds of millions of years. They are characterized by extreme longevity as well as delayed maturity and long reproductive life. Furthermore, their shell is a noteworthy adaptation mechanism, allowing for their survival in both aquatic and terrestrial environments. However, today, more than 50% of 360 ascertained living species of turtles and tortoises are categorized as endangered or vulnerable by the International Union for Conservation of Nature. In more detail, 127 species out of 360 are considered as endangered or critically endangered, while seven species and ten taxa were declared extinct in the last 280 years and three of them in the past decades. This feature shows how chelonians account among the organisms with the highest risk degree of extinction when compared to other reptiles. Moreover, many other taxa, namely, marine and freshwater turtles, could be predicted to become extinct during the present century [1,2]. The factors driving this recent, dramatic, and faster decline of chelonians, as well as their extinction risks, are thought to be related to several different factors: habitat loss; degradation and fragmentation represent a major threat for biodiversity, causing the modification of habitats for agriculture; wetland loss; and human population pressure on the landscape. Climate change contributes to the exacerbation of these factors, leading to environmental desertification; these alterations could involve lower tortoise egg production as well, while sea level rise would reduce the number of possible suitable nesting sites for marine turtles. The introduction of invasive animal species acting as food competitors and predators represents a further threat, while exotic plants—mostly grasses—along with climate change, would enhance the risk of fires. The consumption of turtles, tortoises, and turtles’ eggs for human nutrition has been another impacting threat for the last 400.000 years and could have led to the extinction of large tortoise species since 17th century. The medicinal use of turtles and tortoises is also very impactful, especially in Asia. Lastly, many chelonian species are globally collected for international illegal pet trade practices [2].

In such scenarios, infectious agents could play a capital role in the decline of these populations. Among them, fungal infections may represent a main threat to ecosystem health [3].

Infectious diseases are among the top five reasons for terrestrial species’ [4] and sea turtles’ extinction; infectious hazards can also comprehend fungal affections. Furthermore, the emergence of pathogens could be driven by environmental factors, as hypothesized for the herpes virus, which is responsible for fibropapillomatosis in sea turtles [5] or infection by Ranavirus. However, available data, mostly regarding the sanitary status of sea turtles, are fragmentary, concerning the difficult-to-retrieve diseased animals and carcasses in the wild. Thus, in a lot of cases, it is difficult to correlate morbidity rate with mortality [6].

The present narrative review aimed to gather the main information available from the literature dealing with fungal diseases of Chelonians.

Mycoses in Chelonians have different features; these animals can be affected by fungi causing shell, tegumental, and/or systemic disease, which can be considered as conventional agents. However, in recent years, new emerging fungal pathogens have affected sea and freshwater turtles and their eggs, inducing not previously reported on clinical outcomes that are able to significantly reduce turtle populations worldwide [3,7]. Microsporidiosis in these species has been recently reported on, and literature records are referred to also.

## 2. Mycoses of Chelonians by Conventional Agents

### 2.1. Conventional Mycoses

Conventional mycoses of reptiles have been deeply treated by Austwick and Keymer, 1980, Kotska et al., 1997, Jacobson et al., 2000, and Parè, 2014 [8,9,10,11]; however, further records have been added since these studies were performed. The diseases were caused by environmental fungi, belonging to a wide range of different taxa. Some reports have not been accomplished by mycologists; thus, culture isolation of causative agents has not always been carried out: for this reason, it was not possible in all described cases to identify the responsible fungal species.

#### 2.1.1. Dermatomycoses

Turtles and tortoises can be affected by mycoses involving the skin, scale armor, and carapace. Such affections are considered as related to inadequate management, mostly due to deficiencies in housing such as inappropriate temperature and moisture [12], overcrowding, poor diet, and intercurrent bacterial and parasitic affections, [13], while fungal colonization can follow trauma. In tortoises, shell fractures and skin lesions are mostly caused by vehicles, mowers, falls, or predation by dogs, foxes, or other carnivores. Motored aquatic vehicles have been also reported to damage turtle shells. Skin injuries can be induced by intra or interspecies aggression, predators, fishing lines, or fishing nets.

Inflammation, ulceration, and perforation to abscess formation or necrosis of the underlying tissues or granulomas appear as distinctive features of dermatomycoses [13,14,15]. Being reptiles lacking in subcutaneous fat, muscles and bones may be rapidly involved [13,14], especially in chelonians [15].

##### Shell Mycoses

Shell mycoses have been reported to involve only the carapace or to colonize skin layers.

Most necroses of carapaces would have mycotic etiologies [16]. In tortoises, the lamellae of the carapace may become weak or loose, while small impressions can develop, leading to a fracture of the shell [9,15].

The involvement of shells alone has been mostly observed in tortoises and seldom in turtles [16]. The occurrence of yeasts (*Candida* sp., *Cryptococcus* sp. *Geotrichum candidum* and *Rhodotorula* sp.), together with molds (*Absidia, Alternaria, Aspergillus* spp., *Cladosporium, Drechslera, Fusarium, Mucor, Penicillium, Stemphylium* and *Ulocladium*), has been recorded [17,18]. Mucorales have also been reported to be responsible for shell mycosis in several turtles [16].

A qualitative evaluation of fungal flora involved in shell lesions of *Testudo* spp. was investigated in an extensive study carried out in France [19]. Fungal hyphae were observed in 73% of clarified specimens, while 34 different fungal genera were cultured. Fungi belonging to Dematiaceae were observed in about 90% of samples, with *Alternaria, Cladosporium,* and *Aureobasidium* as the most represented genera (47.2%, 41.5%, and 32.1%, respectively). Keratinophilic fungi such as *Chrysosporium* spp., *Myriodontium keratinophilum,* and *Scopulariopsis brevicaulis* were also identified in culture, the latter with positive microscopic features also.

*Fusarium semitectum* (syn. *Fusarium incarnatum, Fusarium pallidoroseum*) was reported to be responsible for necrotizing scute disease in *Gopherus berlandieri* in Texas [20] and for shell mycosis in 15 *Testudo hermanni* living in a community [21]. The aetiological agent was cultured from the soil. All affected subjects recovered after a specific treatment, as well as after removal from the infected environment, thus indicating the role of soil-inhabiting fungi to be responsible for shell colonization in tortoises. *Fusarium solani* was recently cultured by severe erosive and ulcerative lesions in the carapace of *Graptemys ouachitensis,* involving skin also [22].

A novel species within Onygenales, *Aphanoascella galapagosensis,* was isolated from the diseased carapace of a *Chelonoidis nigra microphyes*. The mold was morphologically described, resulting in differences from other genera within the order. These findings were corroborated by the results of the D1/D2 sequences, demonstrating that this fungal isolate represented a new lineage within Onygenales. Molecular data obtained in this work were compared with the neighbor genus *Aphanoascus* and other related genera also [23].

Melanin-pigmented (dematiaceous) fungi responsible for phaeohyphomycoses were also reported as a cause for shell mycoses. *Exophiala oligosperma* was identified by molecular techniques in a *Geochelone gigantea* infection, involving deep bones [24]. A further recent finding described the identification and isolation of *Alternaria* sp. from a single case of shell mycosis in *T. hermanni*) [25]. Melanized fungal elements were easily noticeable in diaphanized samples (Figure 1).

Shell mycosis due to *Purpureocillium lilacinum* (formerly *Paecilomyces lilacinus*) [26] was observed in *Carettacheylis insculpta* [27], and *Fusarium* spp. were isolated from shell lesions in *C. caretta* [28]. *Fusarium solani* was recently isolated from *C. caretta* with shell and skin involvement [29].

Hatchling Florida soft-shell turtles (*Apalone ferox*) were reported to die from mucormycosis, involving both skin and shell [30], while similar clinical outcomes caused by *Trichosporon cutaneum* have been described in tortoises [31]. Considering that *Trichosporon jirovecii* was considered a synonym of this yeast species until 1992 [32], the involvement of this latter fungal agent cannot be excluded.

##### Skin Mycoses

Turtles and tortoises seem to be prone to skin mycoses involving keratin layers of stratum corneum only, with no shell involvement. Mycotic granulomas, possibly caused by *Aspergillus* sp., were described in *Sternotherus odoratus* [33], while *Mucor* was isolated from skin lesions in *Clemmys insculpta* [10].

*P. lilacinum* was cultured and identified from the shells and skin of young *Trionyx sinensis* affected by white-spot disease [34].

*Fusarium solani* was identified as the main responsible agent for such clinical features. It was mostly identified in *C. caretta* [27,28,35,36,37] and in *Lepidochelys kempii* [38]. In some cases, the same mycotic species was isolated from the environment (sand, water) also [35,36]. The role as opportunistic pathogen of *F. solani* in lesions in turtles was assessed in 1999 by Cabanes [36,39]; the author demonstrated that the banding patterns yielded by an RAPD assay of clinical and environmental samples were markedly differed from those of isolates from several culture collections. Then, *F. solani* was recognized as a monophyletic species complex (FSSC), including more than 60 phylogenetic species [3,40,41,42]. The isolates from sea turtles have been identified as *Fusarium falciforme, Fusarium keratoplasticum,* and haplotypes 9 and 12 [42,43]. Furthermore, in a recent extensive study on *C. caretta* with superficial lesions, *F. keratoplasticum, Fusarium oxysporum, Fusarium brachygibbosum,* along with haplotypes 9, 12, and 27 were identified, while animals with no disease were colonized by haplotypes 9 and 12 only [27]. Similarly, *F. falciforme*, *F. keratoplasticum*, and *Fusarium crassum* were isolated from stranded post-hatchling *C. caretta* in South Africa [44].

In sea turtles, the occurrence of *Fusarium* spp. was suggested to be related both to the ability of these fungi to germinate within water [45] and to microclimatic factors that provoked cold stunning in turtles [37].

Regarding tortoises, *T. jirovecii* was identified from neck skin lesions in *T. hermanni,* recovering after specific antifungal treatment [46].

#### 2.1.2. Deep and Systemic Mycoses

Deep, mostly pulmonary mycosis was reported to be more frequent in tortoises rather than in turtles [16], although further records in aquatic species are present in the literature. In these latter species, environmental fungi were described as able to colonize lesions, i.e., injury caused by the ingestion of a fishhook, as turtles are occasional scavengers. Poikilotherms are prone to adverse environmental factors. Fungal infections depend on poor cellular immunity as well as on adverse environmental conditions (overcrowding, poor water quality, drop in body temperature) or on intercurrent infections. However, the primary causes have frequently not been recognized, mostly in wildlife lacking an anamnesis, the records being made from dying subjects or carcasses.

To the best of our knowledge, the first record of generalized aspergillosis in *Hydraspis hilarii* dates back to 1934 [47].

Entomopathogenic fungi such as *P. lilacinum*, *Isaria fumosorosea* (formerly *Paecilomyces fumosoroseus*), and *Beauveria bassiana* were involved in deep mycoses in *Aldabra tortoises*, *Trachemys scripta*, *C. caretta* [48,49,50,51,52], and in systemic colonization in *Eretmochelys imbricata* and *C. insculpta* [26,53]. *Beauveria bassiana*, *Beauveria brongniartii*, *Metarhizium anisopliae*, *Metarhizium robertsii*, and a novel *Metarhizium* sp. were recently isolated from the lungs of *C. caretta*, *Gopherus polyphemus*, *Chelonia mydas*, and coelomitis in *E. imbricata* [54]. *Paecilomyces* spp. were identified as etiological agents in pneumonitis in *C. mydas* alone [55] or associated with *Sporothrichium* sp. and *Cladosporium* sp. [56]. The phytopathogen *Colletotrichum acutatum* was identified in *L. kempii* with signs of disseminated mycosis [57].

*Penicillium griseofulvum* was involved in systemic mycosis in *Megalochelys (Geochelone) gigantea* [58], *Trichophyton* sp. in *Lepidochelys olivacea* [59], as well as *Fusarium* sp. in severe pneumonia in *L. kempii* [60], while, presumably, *Fusarium* sp. and *Mucor* sp. were demonstrated in *Emys orbicularis* affected by systemic disease [61]. A disseminated phaeohyphomycosis was described in *Geochelone nigra* [62], the dematiacfungi *Cladosporium cladosporioides* and *Alternaria arborescens* were identified in *C. caretta* affected by nephritis, and *Ampelomyces* sp. DNA was recovered from granulomatous lesions of peritoneum of the same subject [63], as well *Veronaea botryosa* from the the obstructive tracheitis in 3 *C. mydas* [64].

Yeasts have been also identified, seeming to behave as opportunistic agents. *Candida albicans* was considered to be responsible for stomatitis, hyperemic or ulcerative gastritis, and enteritis in Chelonians [13,30] and was involved in severe unilateral pneumonia in *Testudo graeca* [65] and in *C. caretta* [66]. Gastrointestinal infection by *Candida* sp. was observed in *G. gigantea* [67]. Other non-*albicans Candida* species (mostly *Candida tropicalis*) were reported to be responsible for stomatitis, otitis, pneumonia, enteritis, and nephritis in tortoises and terrapins [13,30,68], while histological evidence showed that *T. cutaneum* was recovered from cases of stomatitis, enteritis, pneumonia, and granulomatous inflammation in multiple tissues [9], and *Geotrichum candidum* occurred in *Geochelone elephantopus* affected by dermatitis and nephritis [69]. *Candida krusei* was reported to be responsible for fatal, systemic mycosis in *Aldabrachelys gigantea* [70]. *Candida palmioleophila,* associated with bacteria, was recently identified in a *C. mydas* showing systemic disease and candidaemia [71].

Table 1 summarizes the main features of the infections described in the literature.

## 3. Mycoses of Chelonians by Unconventional Agents

### 3.1. Emerging Fungal Pathogens

In recent years, we have attempted to stop several fungal emergencies threatening several animal populations, such as amphibian chytridiomycosis, white nose syndrome in bats, snake fungal disease in wildlife, resulting in serious effects on the population level.

Similarly, sea turtle eggs fusariosis (STEF) and infections by *Emydomyces testavorans* have been reported as fungal emergencies in Testudines.

#### 3.1.1. Sea Turtles’ Egg Fusariosis

This disease is reported to be responsible for low egg hatching from nests both in the wild and in hatcheries [72], mostly consisting of fungal infection by *Fusarium* spp., a fungal genus with worldwide distribution.

Although *F. solani* has been reported as present in sea turtle eggs [73,74,75], fungal identification relied on phenotypes, making it difficult to know whether the isolated molds belonged to the same species. Only recently have phylogenetic characterizations and comparisons been performed, mostly through the study of nuclear internal transcribed spacer (ITS) regions, allowing researchers to identify *F. falciforme* and *F. keratoplasticum,* both belonging to Clade III of the FSSC [3,76,77]. This complex appeared to be strongly involved in veterinary mycology [43], comprehending the etiological agents of about two thirds of all reported fusarioses [78] and acting as a true pathogen, fulfilling Koch’s postulations [76]. It represents a threat both for sea and freshwater turtles’ eggs, being possibly responsible for freshwater turtle egg fusariosis (FTEF) [79].

These fungal species have been identified in beach sand and are possibly brought by overflows from plumbing systems for human waste colonizing turtles’ nesting sites. *Fusarium falciforme* and *F. keratoplasticum* have been found in such environments [80,81]. Fusaria are environmental molds, whose taxonomy has been consistently revised, causing controversial issues. The species are grouped in complexes, some of which have been proposed as distinct genera, such as *Bifusarium* (for the *Fusarium dimerum* species complex) and *Neocosmospora* (for the *F. solani* species complex). However, they will be treated as belonging to the *Fusarium* genus here. Morphologically, they are characterized by banana-shaped macroconidia and are currently considered as emerging pathogens involved in plant, animal, and human infections. Fusaria are, in fact, found in the underground and aerial parts of plants on decaying vegetal materials. In particular, FSSCS are the fifth plant pathogen among the first ten responsible for human infections in hospital environments. In veterinary medicine, they are referred to as aetiological agents for different clinical presentations, mostly in aquatic animals. These fungi have recently been reviewed in a One Health perspective [82], suggesting a possible relationship between STEF and human fusarioses.

FSSCs have been mostly described in C. caretta, C. mydas, Dermochelys coriaceae, Eretmochelys imbricata, L. olivacea, L. kempi, and Natator depressus [3]. However, eggs of freshwater turtles can be affected also. Fusarium keratoplasticum, as well as other members of the FSSC, have been found in Podocnemis unifilis [79,83]. Furthermore, the invasive alien species Trachemys scripta has recently been suggested as a carrier of FSSCs in Mediterranean freshwater marshes [84].

Sea turtles’ eggs are buried in the sand and in nesting sites, mostly in slimy and silty environments [3], which have suitable conditions for the growth of these fungi, which feed on organic matter deriving from hatched and failed eggs [85]. Affected eggs can show yellow, blue, or red areas on eggshells, but more severe infections are able to develop in gray hyphal mats [76]. Hyphae can produce a network on damaged eggs, colonizing neighboring eggs also [86]. The molds produce enzymes and organic acids that are able to dissolve the shell, invading the embryos [76,87]. However, these fungi can also be present in nests with asymptomatic eggs [76], suggesting that factors other than climatic conditions would play a role in STEF [85]. FSSC members widely occur in terrestrial environments and could reach the oceans by run offs, colonizing floating particles [85]. The role of the ingestion of insoluble organic and inorganic particles and microplastics as substrates for the growth of potentially pathogenic fungi, released with feces by sea turtles, have been discussed [88,89] to elucidate the occurrence of fungal cells in nests. Furthermore, the effect of mycotoxins cannot be ruled out, as well as the protective activity of protease inhibitors produced by some turtles. Another important feature is represented by the identification of microbiomes in *Fusarium*-infected sea turtle eggs in protection against STEF. A preliminary study involving microbial community occurances in the nesting sites of *E. imbricata* in Ecuador, in order to evaluate bacteria with antifungal activity, allowed researchers to determine that Actinobacteria isolated from healthy eggshells, mostly *Streptomyces*, as well *Amycolaptosis*, *Micromonospora,* and *Plantactinospora*, showed in vitro activity against *F. falciforme* [90]. Despite the small sample size of the study, these findings shone a light on the microflora associated with turtle eggs and are corroborated by recent studies on symbiotic bacteria of bobtail squids (*Euprymna scolopes*). This bacterial community produced secondary metabolites that were able to inhibit the overgrowth of fungi and bacteria responsible for egg fouling, with marked antifungal activity against *F. keratoplasticum* [91,92].

As reported above, *Fusarium* is the most involved fungal genus in turtles’ and tortoises’ mycoses. Chelonian species, etiologic agents, and clinical outcomes are reported in Table 2. For details concerning the mere isolation of Fusarium spp. from sea turtles, please refer to Gleason et al. [85].

#### 3.1.2. *Emydomyces testavorans* Infection

*Emydomyces testavorans* is a member of the order Onygenales, recently isolated and characterized by ulcerative skin and shell lesions in several species of freshwater turtles from the USA [93,94]. This disease is considered a threat for the conservation of *Actinemys marmorata* [95] and presents similarities with Septicemic Cutaneous Ulcerative Disease, a bacterial syndrome found in captive turtles. However, *E. testavorans* infection differs from other shell mycoses, the general conditions of animals not being involved [95]. The lesions consist of keratin inclusion cysts, limited by keratinized squamous epithelium; hyperkeratosis, inflammation, and osteonecrosis are observed as shell lesions [96]. Affected animals show shell pitting also. Cysts can expand into celomatic cavity, compressing internal organs.

The disease is often underdiagnosed, especially when external lesions cannot be appreciated [94,97], and the etiological agent, until today, is not recognized as a true pathogen [97], unlike the other emerging Onygenales, *Ophydiomyces* and *Nannizziopsis,* in snakes and chameleons. The ecological situations and the environmental conditions of the areas in which *E. testavorans* infection has been reported on should be further investigated to elucidate the epidemiology of this emergent mycosis. However, seasonal variations, as observed for snake infections, cannot be ruled out [94]. Young *Macrochelys temminckii* can show different clinical signs such as rhinitis, paronychia, nail loss, skin ulcers, plastron ulceration, excessive shedding, and death. Shedding may persist when animal husbandry is ameliorated, and the disease becomes chronic [97].

#### 3.1.3. Microsporidioses

Microsporidia are a group of obligate intracellular single-celled eukariotes, lacking mitochondria, recently included in the Kingdom Fungi [98]. They spread through spores and would infect a huge range of hosts, including reptiles [99]. Microsporidiosis was first reported in four tortoises (*Testudo hermanni boettgeri*) from Germany, with intestinal and respiratory clinical signs. About 20 subjects living together with the 3 examined tortoises showed respiratory and digestive clinical signs, and most of them died within three weeks from the onset of illness. The animals became symptomatic after hibernation and died within early summer. Histopathological findings allowed researchers to detect granulomatous and necrotizing hepatitis and/or pneumonia, containing large amounts of spores, also observed in the small intestine in one animal (Eydner et al., 2017) [100]. Bacteriological cultures reported as positive for Enterobacteriaceae and *Streptococcus* spp. Microsporidial organisms were identified neither at a species nor a genus level. Regardless, the number and location of filament coils allowed the researchers to exclude the occurrence of *Encephalitozoon* spp., described in *Pogona vitticeps* [101,102].

## 4. Treatment of Mycoses in Chelonians

Many reports deal with description of clinical and mycological pictures in wild Chelonians; sea turtles; and, except for epidemics, in cases in captive or pet subjects and stranded animals brought up at rehabilitation centers. Nevertheless, data about specific antimycotic treatments, follow ups, and outcomes are scarce. Animal species, etiological agents, clinical presentations, drug posologies, and outcomes are reported in Table 3.

## 5. Conclusions

Chelonians can be affected by mycoses caused both by conventional and emerging aetiologic agents.

Conventional mycoses in captive and pet reptiles would depend on poor husbandry, the agents being mostly opportunistic pathogens. Other possible causes for wild animals are stressors such as concurrent bacterial infections, parasites, wounds, ingestion of foreign bodies, exposure to environmental contaminants, or being caught in fishing nets. However, some fungal species seem to occur more frequently. *Fusarium* spp. are frequently involved with different clinical features, as previously reported. However, the entomopathogenic mold *P. lilacinum* has been consistently observed in Chelonians’ mycoses, suggesting a tropism of this organism toward reptile tissues. This fungal species has been recently recovered from unhatched eggs in *T. hermanni* associated with bacteria (*Pseudomonas aeruginosa, Bacillus* sp. and *Escherichia coli*) [103]. Furthermore, *P. lilacinum* is considered a primary pulmonary pathogen in green tree pythons [104], and the related species *Purpureocillium lavendulum* has been isolated from lesions in several reptile species [105].

The impact of fungal affection on turtles and tortoises’ health has not been fully understood yet, although some emerging agents such FSSCs have been recognized as real threats to the survival of some aquatic species and has been recently included within pathogens with One Health issues [82]. Furthermore, in the last few years, the role of the FSSC as a primary pathogen has been assessed. This feature has not been recognized for *E. testavorans,* due to its recent identification.

Future work should include a multidisciplinary approach to mycotic diseases in these animal species. In fact, data about the risk management for fungal disease hazards are limited by the scarce number of experts in this area, resulting in difficulty and, consequently, a lack of diagnostic testing in a lot of regional management units [6].

To enhance the study of etiologic agents, with particular paid attention to strain virulence, researchers should endeavor to research the pathogenesis of such infections and the sensitivities of fungal isolates *versus* antimycotic drugs.

## Figures and Tables

**Figure 1 jof-09-00518-f001:**
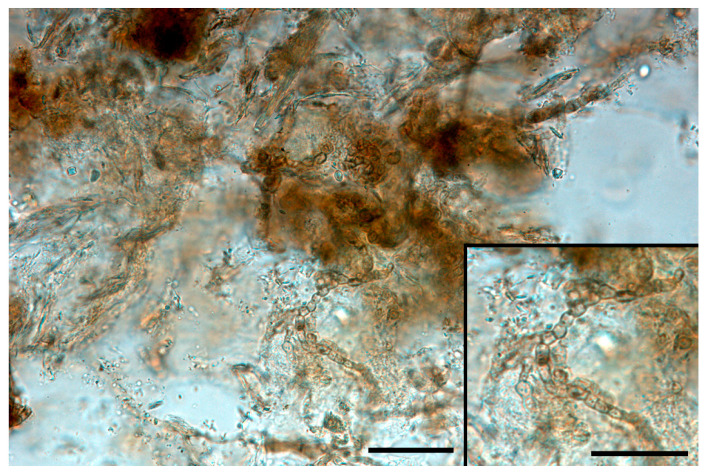
Melanized short hyphae in a diaphanized sample from *Testudo hermanni* carapace (bar = 200 μm). Insert: detail of fungal elements (bar = 100 μm).

**Table 1 jof-09-00518-t001:** Main fungal diseases, with epidemiological patterns, transmission routes, pathogenetic features, and prevention strategies.

Species	Etiologic Agent	Epidemiological Pattern *	Transmission Route §	Pathogenetic Features	Lesions	Prevention Strategies	Ref.
*Gopherus berlendieri*	*Fusarium semitectum*	Group infection ©	Environment	fungal keratinases	superficial		[20]
*Testudo hermanni*	*Fusarium semitectum*	Group infection ©	Environment	fungal keratinases	superficial	moving	[21]
*Graptemys ouachitensis*	*Fusarium solani*	Single case ©		fungal keratinases and poor management	superficial	improved management practices	[22]
*Chelonoidis nigra*	*Aphanoascella galapagosensis*	Single case and group ©			superficial	none	[23]
*Geochelone gigantea*	*Exophiala oligosperma*	Single case ©			superficial	none	[24]
*Testudo hermanni*	*Alternaria sp.*	Single case ©			superficial	none	[25]
*Carettocheylis insculpta*	*Purpureocillium lilacinum*	Group infection (W)		immunosuppression, stress of shipment, poor management	superficial	water changes and good management practices	[26]
*Caretta caretta*	*Fusarium spp*	Group infection © (R)	Environment	stressors, poor management	superficial pneumonia	improved management practices	[27]
*Caretta caretta*	*Fusarium solani*	Group infection © (R)		cold stunning	superficial		[28]
*Apalone ferox*	*Mucorales*	Group infection ©		mixed bacterial infection	superficial		[29]
*Trionyx sinensis*	*Purpureocillium lilacinum*	Group infection ©		poor management	superficial	improved management practices	[34]
*Caretta caretta*	*Fusarium solani*	Single case © (R)	Environment	immunosuppression for trauma, surgery, rehabilitation	superficial	improved management practices	[36]
*Caretta caretta*	*Fusarium solani*	Group infection ©	Environment through wound	fungal keratinases, stress for stranding, and poor management	superficial	improved management practices	[37]
*Lepidochelys kempii*	*Fusarium solani*	Single case © (R)	Through wound	cold stunning, mixed bacterial infection	superficial		[38]
*Caretta caretta*	*FSSC*	Group infection (W) (R)		stranded	superficial	preventing pathogen introduction and diffusion	[44]
*T. hermanni*	*Trichosporon jirovecii*	Single case ©	Through wound	Infection of a skin lesion	superficial	preventing infections of open wounds	[46]
*Caretta caretta*	*Purpureocillium lilacinum*	Single case ©	Water	poor management, introduction of infected hosts	pneumonia	improved management practices	[51]
*Caretta caretta*	*Purpureocillium lilacinum*	Group infection ©		captive stress, nutritional deficits, poor management	superficial pneumonia	improved management practices	[52]
*Lepidochelys kempii*	*Colletotrichum acutatum*	Single case © (R)		Cold stunning			[57]
*Megalocheylis gigantea*	*Penicillium griseofulvum*	Single case ©		stress consequent to burns	systemic		[58]
*Chelonia mydas*	*Veronaea botryosa*	Group infection (W)	Respiratory	stranded animals	obstructive tracheitis		[64]
*Chelonia mydas*	*Candida palmioleophila*	Single case © R		shell fracture	systemic		[71]

Legend: * © captive animals; (W) wild animals; (R) animals introduced in a rehabilitation center. § All the listed fungi are environmental inhabitants, but the captions refer to confirmed occurrences of agents in the environment.

**Table 2 jof-09-00518-t002:** Fusarioses of Chelonians.

Animal Species	Agent	Country	Disease	Reference
*Gopherus berlandieri*	*Fusarium semitectum*	Texas	Necrotizing scute disease	[20]
*Testudo hermani*	*F. semitectum*	Italy	Shell mycosis	[21]
*Graptemys ouachitensis*	*Fusarium solani*	Portugal	Shell mycosis	[57]
*Caretta caretta*	FSSC *	Italy	Skin and shell	[27]
*Caretta caretta*	*F. solani*	Italy	Skin, shell and bone	[28]
*Caretta caretta*	*F. solani*	Spain	Skin	[36]
*Caretta caretta*	*F. solani*	France	Skin and systemic	[37]
*Caretta caretta*	FSSC **	South Africa	Skin	[44]
*Lepidochels kempii*	*F. solani*	USA	Abscess	[38]
*Emys orbicularis*	*Fusarium* sp. and *Mucor*	Serbia	Systemic	[66]
*Caretta caretta*	*F. solani*	Cape Verde	STEF	[76]
*Chelonia mydas*	FSSC	Australia, Ecuador	STEF	[3]
*Eretmochelys imbricata*	FSSC	Ecuador	STEF	[3]
*Lepidochelys olivacea*	FSSC	Ecuador	STEF	[3]
*Dermochelys coriacea*	FSSC	Colombia, Costa Rica	STEF	[3]
*Natator depressus*	FSSC	Australia	STEF	[3]
*Podocnemis unifilis*	FSSC	Ecuador	FTEF	[79]
*Podocnemis unifilis*	FSSC	Ecuador	FTEF	[83]

Legend: FSSC: *Fusarium solani* species complex; STEF: sea turtle egg fusariosis; FTEF: freshwater turtles. * *F. keratoplasticum*, *Fusarium oxysporum*, *Fusarium brachygibbosum,* and haplotypes 9, 12, and 27. ** *F. falciforme, F. keratoplasticum,* and *Fusarium crassum.*

**Table 3 jof-09-00518-t003:** Mycoses of chelonians submitted to specific antimycotic treatments.

Animal Species	Agent	Disease	Antimycotic Drugs	Outcome	Reference
*Aldabrachelys gigantea*	*P. lilacinum*	systemic	Ketoconazole 10 mg/kg PO	death	[48]
*Aldabrachelys gigantea*	*C. krusei*	systemic	Itraconazole 5 mg/kg/die PO	death	[70]
*Lepidochelys kempi*	*Co. acutatum*	systemic	Fluconazole 0.75 mg/kg SC/48 h, then Itraconazole 5 mg/kg/die PO	death	[57]
*Caretta caretta*	*F. solani*	shell	Topical 10% iodine in alcohol and topical ketoconazole	recovery	[36]
*Caretta caretta*	*F. solani*	Skin and systemic	Posaconazole 0.2 mg/kg/48 h PO	recovery by 33%	[37]
*Testudo graeca*	*C. albicans*	Pneumonia	Amphotericin B 0.1 mg/kg/die intrapulmonary	recovery	[65]
*A. marmarota*	*E. testavorans*	Skin shell	Topical iodine, terbinafine	recovery	[97]
*Carettochelys insculpta*	*P. lilacinum*	shell	Malachite green and formaldehyde dips, itraconazole 10 mg/kg/48 h PO	recovery	[25]
*T. hermanni*	*F. semitectum*	shell	Povidone iodine and Iruxol ointment/daily	recovery	[21]
*T. hermanni*	*T. jirovecii*	skin	Povidone iodine and Iruxol ointment/daily	recovery	[45]
*T. hermanni*	*Alternaria* sp	shell	Topical 10% iodine in alcohol	recovery	[24]

## Data Availability

Data sharing not applicable.

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
