# Peer review of "Mycotic Diseases in Chelonians"

_jof, 2023, doi:10.3390/jof9050518_

Round 1

Reviewer 1 Report

The aim of the present narrative review was to gather main information about the literature, dealing with fungal diseases of Chelonians. This paper is very helpful for a comprehensive understanding of fungal diseases in Chelonians. However, the overall writing of the article is relatively simple, with more emphasis on listing the fungal diseases that have already occurred. I suggest that the author make a table of the fungal diseases that have occurred, and at the same time, focus on summarizing the epidemic patterns, transmission routes, and prevention strategies of these fungal diseases. A detailed analysis of the pathogenic characteristics of fungal diseases and future research priorities should also be conducted. Expanding these contents will help the author better understand fungal diseases of Chelonians.

Author Response

Reviewer 1

The Authors would thank the reviewer for his (her) revision, that helped to increase the scientific value of the whole text.

An extensive editing of English language and style was provided.

A table of the fungal diseases that have occurred was added the text, as required.

Epidemic patterns, transmission routes and prevention strategies of these infections were summarized, and an analysis of the pathogenic characteristics and future research priorities about this topic were included, too.

Reviewer 2 Report

The reviewed article provides a detailed overview of the various mycotic diseases that can affect chelonians, including the causal agents, lesions, and treatments.

I have just a few more minor comments for the authors to address, which I hope they will find useful in improving their manuscript:

Line 72: correct the word “and” in the text “Kotska et al, 1997 and Jacobson et al., 2000, Parè, 2014”. Should be: Kotska et al, 1997, Jacobson et al., 2000 AND Parè, 2014.

Line 109: correct the word “acute”

Line 114: add a space between “involving” and “skin”.

Line 122 and 190: I suggest the use of dematiaceous or melanin-pigmented instead of dark.

Line 199: add space between the reference and the word “and”

Line 201: Please, explain the term polygranulomatosis. Granulomatosis usually refers to a specific immune-mediated and non-infectious condition associated with polyangiitis. Granulomas associated with infectious diseases, like E. coli in chickens, could receive a specific name as Coligranuloma. I suggest changing the term to granulomatous inflammation in multiple tissues.

Lines 293 to 296 and table 1: I suggest changing the position of the text after section 3.1.1 and above section 3.1.2.

References: Please, review the references, especially 98 – 103.

Author Response

Reviewer 2

The Authors would thank the reviewer for his (her) revision, that helped to increase the scientific value of the whole text.

An editing of English language and style was provided.

Further comments and observations were addressed, as better specified below:

Line 72: the reference citation was adjusted

 Line 109: the word was corrected.

Line 114: the space was added.

Lines 122 and 190: the suggested words were inserted in the text.

Line 201: the term polygranulomatosis was explained.

Lines 293 to 296 and Table 1: the suggested changes were taken up in the text, as suggested.

 References were conformed to the Journal’s style, as requested.

Round 2

Reviewer 1 Report

It is ok.